# A unique binding mode of Nek2A to the APC/C allows its ubiquitination during prometaphase

Claudio Alfieri[*,†] iD, Thomas Tischer iD & David Barford iD

## Abstract

The anaphase-promoting complex (APC/C) is the key E3 ubiquitin ligase which directs mitotic progression and exit by catalysing the sequential ubiquitination of specific substrates. The activity of the APC/C in mitosis is restrained by the spindle assembly checkpoint (SAC), which coordinates chromosome segregation with the assembly of the mitotic spindle. The SAC effector is the mitotic checkpoint complex (MCC), which binds and inhibits the APC/C. It is incompletely understood how the APC/C switches substrate specificity in a cell cycle-specific manner. For instance, it is unclear how in prometaphase, when APC/C activity towards cyclin B and securin is repressed by the MCC, the kinase Nek2A is ubiquitinated. Here, we combine biochemical and structural analysis with functional studies in cells to show that Nek2A is a conformational-specific binder of the APC/C–MCC complex (APC/C$^{MCC}$) and that, in contrast to cyclin A, Nek2A can be ubiquitinated efficiently by the APC/C in conjunction with both the E2 enzymes UbcH10 and UbcH5. We propose that these special features of Nek2A allow its prometaphase-specific ubiquitination.

**Keywords** anaphase-promoting complex; cell cycle; cryo-EM; E3 ligase; spindle assembly checkpoint
**Subject Categories** Cell Cycle; Post-translational Modifications & Proteolysis; Structural Biology

See also: **J Nilsson** (June 2020)

## Introduction

Successful mitosis relies on the ordered degradation of specific cell cycle regulators. Most of these regulators in mitosis are ubiquitinated and targeted for proteasomal degradation by the large multi-subunit E3 ubiquitin ligase, the anaphase-promoting complex/cyclosome (APC/C) [1–4]. APC/C activity directs sister chromatid separation at the metaphase-to-anaphase transition, mitotic exit and the establishment and maintenance of G1. APC/C-dependent ubiquitin chain initiation is promoted by the two E2 enzymes, UbcH10 and UbcH5 [5–7].

One fundamental question is how the APC/C promotes the ubiquitination of distinct substrates in an ordered fashion throughout the cell cycle. The substrate recognition of the APC/C depends on its association with the coactivator subunits Cdc20 and Cdh1, which bind the APC/C in mitosis and during mitotic exit, respectively, and recognize degron sequences on APC/C substrates [8]. Coactivators also promote an active conformation of the APC/C catalytic module, which allows for E2 binding and ubiquitin chain initiation [9]. The coactivator-dependent degrons include the D- and KEN-boxes, and ABBA motif [10], which are recognized by distinct binding sites on the coactivator-WD40 domain. Recognition of the D-box also requires the APC/C subunit Apc10 [11].

APC/C activity in prometaphase is restrained by the mitotic checkpoint complex (MCC), which is the effector of the spindle assembly checkpoint (SAC) [12–15]. The SAC coordinates sister chromosome segregation with the achievement of correct chromosomal attachments to the mitotic spindle [16]. The MCC is composed of Cdc20, Mad2, BubR1 and Bub3. The MCC and APC/C$^{Cdc20}$ associate to form the APC/C$^{MCC}$ complex, which therefore contains two molecules of Cdc20: one is the APC/C coactivator (Cdc20$^{APC/C}$) and the other is Cdc20 of the MCC (Cdc20$^{MCC}$) [17]. BubR1 uses pseudo-degron sequences to block coactivator-dependent substrate recognition [13–15]. Cyclin A ubiquitination starts early in mitosis, and this requires a direct competition of the cyclin A degrons with the pseudo-degron sequences in BubR1. This allows cyclin A recognition and ubiquitination by the APC/C$^{MCC}$ [18,19].

Importantly, APC/C$^{MCC}$ features two main conformations: APC/C$^{MCC}$ closed (APC/C$^{MCC-closed}$) and APC/C$^{MCC}$ open (APC/C$^{MCC-open}$) [14,15]. The transition between closed and open states requires the Apc15 subunit [14,15,20–22], which rearranges the MCC binding site on the APC/C platform module [14]. In APC/C$^{MCC-closed}$, both catalysis and substrate recognition are inhibited. In APC/C$^{MCC-open}$, catalytic inhibition is relieved, and upon E2 binding, ubiquitination of MCC subunits is promoted. Recent data show that cyclin A binding to APC/C$^{MCC}$ promotes APC/C$^{MCC-open}$, which allows cyclin A ubiquitination [18].

The protein kinase Nek2A is required for proper formation of the mitotic spindle, and like cyclin A, it is an early substrate of the APC/C in mitosis [23–25]. Elevated expression of Nek2A in some cancer types leads to multi-nucleated cells and aneuploidy [26]. Prior work has shown that its recognition by the APC/C

MRC Laboratory of Molecular Biology, Cambridge, UK
*Corresponding author. Tel: +44 20715 35087; E-mail: claudio.alfieri@icr.ac.uk
†Present address: Institute of Cancer Research, London, UK
[The copyright line has been changed on 23 July, after first online publication.]

depends on both a leucine zipper, which mediates Nek2A dimerization, and an extreme C-terminal motif comprising a Met-Arg dipeptide, termed the MR tail [24]. The MR tail shares similarity with the IR (Ile-Arg)-tail motifs required by coactivators and Apc10 to directly bind the APC/C [27]. It is unknown how the MR tail is recognized by the APC/C at the molecular level, and how this directs Nek2A ubiquitination when the APC/C is inhibited by the MCC.

To study this system, we combined biochemical reconstitution and analysis with cryo-EM structural studies and cell-based functional studies. We show that Nek2A is selectively bound by the APC/C$^{MCC}$ open conformation, allowing its efficient ubiquitination by APC/C$^{MCC}$. Nek2A is ubiquitinated by the APC/C in conjunction with either UbcH10 or UbcH5. This contrasts with cyclin A, where only UbcH10, and not UbcH5, can pair with the APC/C for its efficient ubiquitination. We suggest that these unique properties of Nek2A allow its ubiquitination during prometaphase.

# Results and Discussion

### The MR tail is the main contributor for Nek2A binding to the APC/C and for Nek2A ubiquitination by APC/C$^{Cdc20}$

To understand the molecular mechanism of Nek2A recognition by APC/C$^{Cdc20}$, which allows for its timely degradation, we assessed the contribution of the characterized Nek2A degrons for its ubiquitination *in vitro* (Fig 1A and B, Source data Blots). Mutating the coactivator-dependent KEN-box degron, which plays a role in promoting Nek2A degradation in anaphase [23–25], only mildly affects Nek2A ubiquitination (Fig 1B). The combined mutation of both the KEN- and D-boxes does not further affect Nek2A ubiquitination levels (Fig 1B). By contrast, deletion of the C-terminal MR tail, which is essential for Nek2A degradation in prometaphase [24,25], strongly impairs ubiquitination of Nek2A by APC/C$^{Cdc20}$ (Fig 1B). Combined mutation of all Nek2A degrons completely abolishes Nek2A ubiquitination (Fig 1B, lane 5). These results indicate that the main contributor for efficient ubiquitination of Nek2A by APC/C$^{Cdc20}$ is the MR tail. This agrees with previous studies showing that in cells, D-box and KEN-box mutants only mildly affect Nek2A degradation kinetics, in contrast to the strong effects of disrupting the MR tail [23–25]. Despite the non-essential roles of the D- and KEN-boxes, Nek2A ubiquitination by the recombinant APC/C is Cdc20-dependent, confirming previous studies (Fig 1C) [28].

Consistent with recent studies [29], we found that in analytical size-exclusion chromatography experiments, Nek2A binds the APC/C in the absence of coactivator and that this binding is strictly dependent on the MR tail (Figs 1D and EV1A). Our data are in agreement with previous binding experiments which showed that mutations of the MR tail disrupt Nek2A binding to APC/C$^{Cdc20}$ [24]. Moreover, we found that in contrast to D-box and KEN-box-dependent substrates such as cyclin A, Nek2A does not stabilize Cdc20 binding to the APC/C (Figs 1E and EV1B). Also, Nek2A binding to the APC/C$^{Cdc20}$ is not disrupted by cyclin A, which occupies the D-box, KEN-box and ABBA motif binding sites on the coactivator (Figs 1E and EV1B). These data suggest that the MR tail is the main contributor of Nek2A binding to APC/C$^{Cdc20}$, with a less significant contribution from the D- and KEN-boxes.

### Structure of the APC/C−Nek2A complex

To define the interactions between the Nek2A MR tail and the APC/C at the molecular level, we determined the cryo-EM structure of the APC/C−Nek2A complex at 3.8 Å resolution (Fig EV2A–D and Table EV1). Prior work suggested that the Nek2A binding site on the APC/C involves the Apc8 subunit [24]. Similar to other canonical TPR-containing subunits of the APC/C, Apc8 is a homodimer. The likely subunit that would engage Nek2A is Apc8A. Apc8B is the binding site for the coactivator C-box motif, which is essential for APC/C catalytic activation and substrate ubiquitination [9,28] (Fig 1C). 3D classification of the EM data focussing on Apc8A (Fig EV3A) allowed the identification of two subpopulations of particles in which the internal groove of Apc8A was either occupied or empty (Fig EV3B and C). Strikingly, 3D refinement of the former particles showed an additional density feature consistent with an MR dipeptide (Figs 1F and EV3B). The binding mode of the Nek2A MR tail is virtually identical to the IR tail of Cdc20 of the MCC (Cdc20$^{MCC}$) in the APC/C$^{MCC}$ cryo-EM reconstruction [14], which after reprocessing was refined to a resolution of 3.8 Å (Figs 1G, EV2E, EV4 and 5). Arg445 of Nek2A points towards Apc8 Asn339 (Fig 1F). Mutating Apc8 Asn339 reduces Nek2A binding to the mitotic APC/C [24].

Nek2A is a dimer due to the presence of a leucine zipper (LZ) spanning residues 304-320 (Fig 1A). Because dimerization through its LZ is essential for Nek2A degradation kinetics [24], we hypothesized that a second MR tail binding site exists on the APC/C. Careful inspection of our APC/C cryo-EM reconstruction allowed the identification of an extra density feature corresponding to a globular mass of about 10 kDa located at the interface of the Apc4$^{WD40}$ domain and the Apc2 cullin repeat 3 (Apc2$^{CR3}$) (Figs 2A and EV3D). Focussed 3D classification and refinement of this region allowed the separation of particles containing this density from particles without (Fig EV3E). Refining the first class yielded a better-defined extra density that could be unambiguously assigned to the Apc2$^{WHB}$ domain (Figs 2A and EV3F). Further 3D classification of these data (Fig EV3G) generated a reconstruction containing a defined density wedged into a pocket formed by Apc2$^{WHB}$, Apc2$^{CR3}$ and Apc4$^{WD40}$, into which we fitted the additional MR tail dipeptide (Figs 2A and EV3H). The MR-binding pocket is formed from Asn392, Glu395 and Gln806 of Apc2 and Arg48, Ser51 and His53 of Apc4 (Fig 2B). The interaction between Gln806 of Apc2$^{WHB}$ and Arg of the Nek2A MR tail may explain how Nek2A induces ordering of Apc2$^{WHB}$, a domain of Apc2 which is highly mobile and only visible in APC/C complexes in the presence of either UbcH10 [30] or the MCC [14,15].

Mutation of Apc2 residues Asn392 and Glu395 to alanine had only a mild effect on Nek2A ubiquitination (APC/C$^{2m}$, Fig 2C). Combining APC/C$^{2m}$ with mutations of residues of the MR-binding site of Apc4 (Arg48, Ser51 and His53) essentially abolished Nek2A ubiquitination by APC/C$^{Cdc20}$ (APC/C$^{2/4m}$, Fig 2C). Importantly, these mutations do not impair ubiquitination of the well-characterized APC/C$^{Cdc20}$ substrate securin, whose recognition by the APC/C depends on the coactivator subunit and Apc10 (Fig 2D).

In conclusion, the structure of the APC/C−Nek2A complex shows that recognition of Nek2A involves two MR tail binding sites on the APC/C. MR pocket 1 is the previously characterized site for the Cdc20$^{MCC}$ IR tail on Apc8A [14,15], which is the structurally equivalent site on Apc8A as the C-box binding site on Apc8B [27].

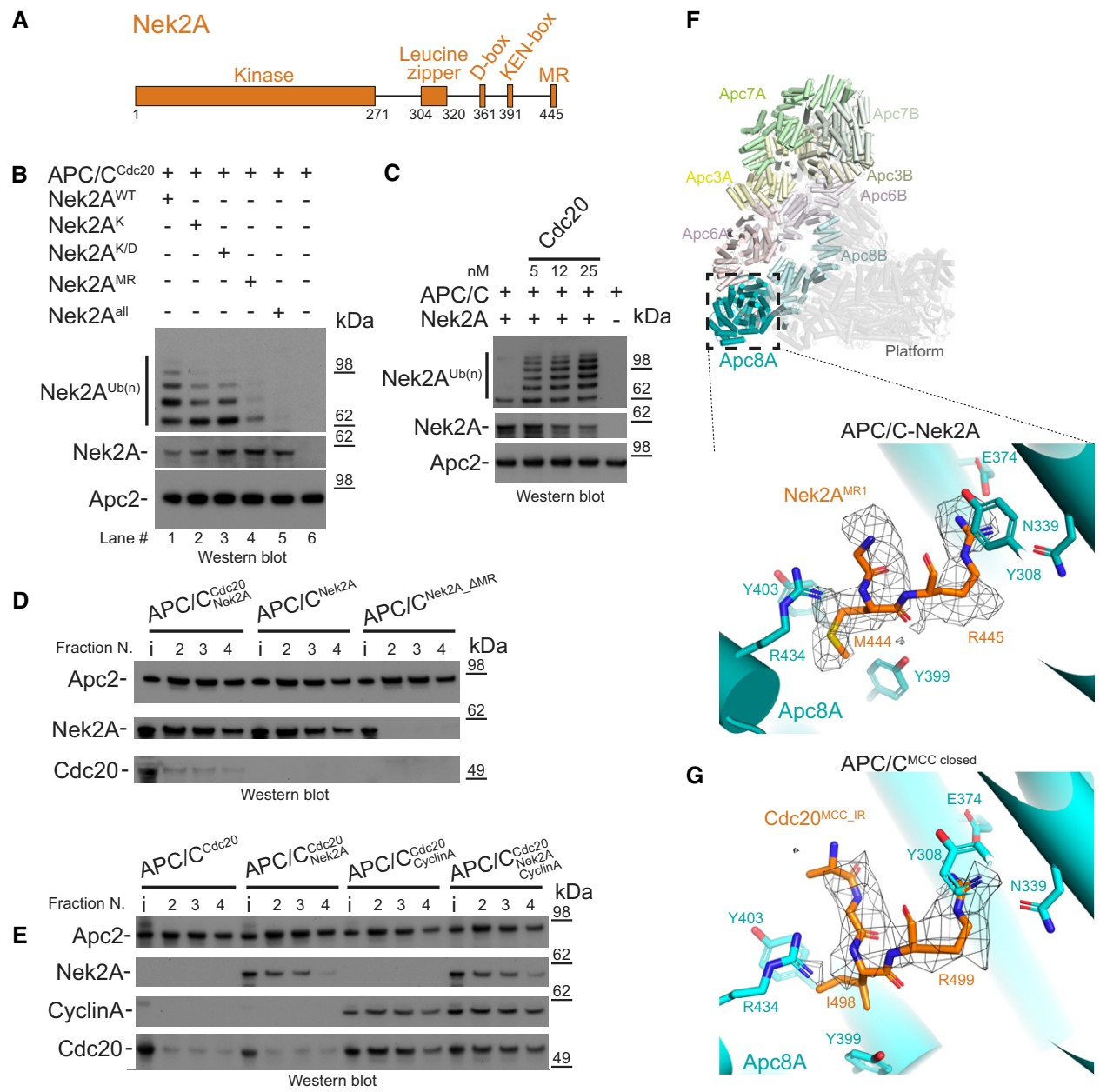

**Figure 1. Contribution of Nek2A degrons in binding and ubiquitination by the APC/C and Nek2A MR tail binding site on Apc8.**

A  Schematic representation of the domain and degron architecture of Nek2A.

B  The MR tail is the main contributor for Nek2A ubiquitination. Ubiquitination reactions performed with APC/C$^{Cdc20}$ for Nek2A wild-type (Nek2A$^{WT}$) and degron mutants. Nek2A$^K$ is the KEN-box mutant (391KEN393/KAA), Nek2A$^{K/D}$ is the (KEN- and D-box mutant (391KEN393/KAA, 361RKFL364/AKFA), and Nek2A$^{MR}$ is the 443ΔMR mutant.

C  The APC/C requires the Cdc20 coactivator subunit to ubiquitinate Nek2A. Nek2A ubiquitination reactions performed with the APC/C in the absence and presence of increasing concentrations of Cdc20 using the E2 UbcH10.

D  The MR tail is essential for Nek2A binding to the APC/C, whereas in contrast Cdc20 is dispensable. Size-exclusion chromatography peak fractions of APC/C complexes with Nek2A. The input material (13% of total) is indicated (i). Peak fractions are numbered in respect of the chromatograms in Fig EV1.

E  Nek2A binding does not stabilize binding of the coactivator subunit Cdc20 to the APC/C and does not compete with cyclin A for binding to the APC/C. Size-exclusion chromatography peak fractions of APC/C complexes with Nek2A.

F  Top: Cylinder representation of the APC/C–Nek2A complex structure. The TPR lobe subunits are highlighted: Apc8 (cyan), Apc6 (pink), Apc3 (yellow) and Apc7 (green). The platform subunits and the small subunits from the TPR lobe are shown in transparency. Bottom: cryo-EM density of the Nek2A$^{MR}$ tail dipeptide and details of the Nek2A$^{MR}$ tail (orange) binding site on Apc8A.

G  Details of the IR tail of Cdc20$^{MCC}$ (orange) bound to Apc8A in APC/C$^{MCC-closed}$. This is the same site as the Nek2A$^{MR}$ tail binding site shown above. Shown is the cryo-EM density for the Cdc20$^{MCC}$ IR tail and contacting residues of Apc8A.

Source data are available online for this figure.

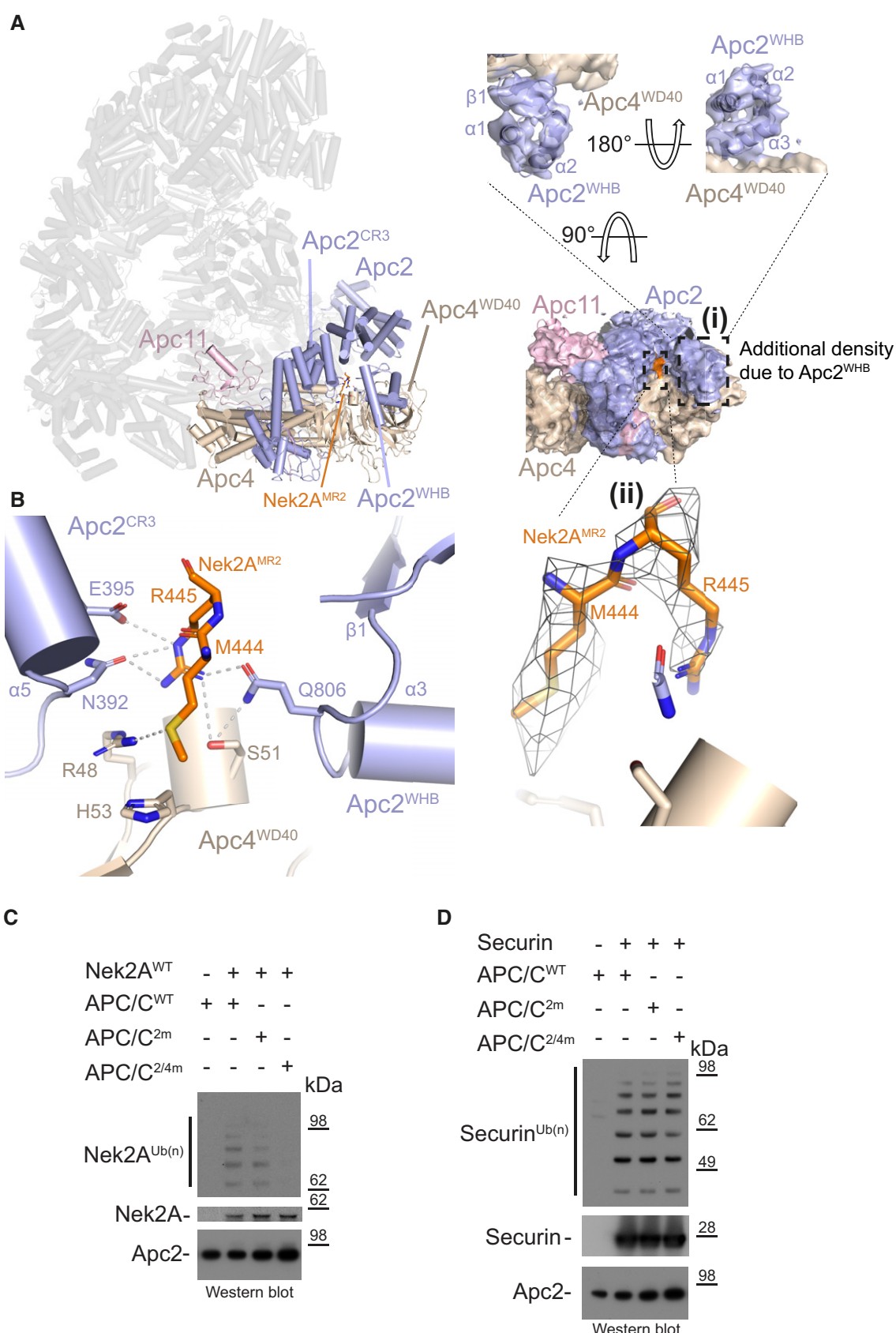

**Figure 2.**

**Figure 2. Nek2A MR tail binding site on Apc2 and Apc4.**

A   Left: Cylinder representation of the APC/C–Nek2A complex structure. Apc2 (light blue) and Apc4 (light brown) are highlighted, and the rest of the APC/C is shown in transparency. (i) Two different views of cryo-EM density for the Apc2 and Apc4 subunits. Secondary structure elements of the Apc2$^{WHB}$ domain are highlighted. (ii) Cryo-EM density for the second MR tail of Nek2A.

B   Details of the second MR tail binding site formed by the Apc2 (including the repositioned Apc2$^{WHB}$) and Apc4 subunits. Nek2A$^{MR2}$ is shown in orange.

C   Nek2A ubiquitination reactions performed with either APC/C$^{Cdc20}$ wild type (APC/C$^{WT}$) or mutants. APC/C$^{2m}$ is the N392A/E395A mutant of Apc2. APC/C$^{2/4m}$ is the N392A/E395A/R48A/H53A/S51A mutant of Apc2 and Apc4.

D   Securin ubiquitination reactions performed with either APC/C$^{Cdc20}$ wild type (APC/C$^{WT}$) or mutants.

Source data are available online for this figure.

MR pocket 2, on the other hand, is a newly identified site at the interface between the Apc2$^{WHB}$, Apc2$^{CR3}$ and Apc4$^{WD40}$ domains.

## Molecular recognition of Nek2A by the APC/C$^{MCC}$ complex

Nek2A is mainly ubiquitinated and degraded in early mitosis. Similar to cyclin A, Nek2A degradation starts in prometaphase when APC/C$^{Cdc20}$ activity is repressed by the MCC [23]. Previous studies had shown that in prometaphase, Nek2A copurifies with both apo-APC/C and APC/C$^{MCC}$ complexes [24,25]. Apo-APC/C cannot be responsible for Nek2A degradation in prometaphase because in the absence of coactivator, the catalytic module of the APC/C cannot bind the E2 enzyme (either UbcH10 or UbcH5) required to initiate substrate ubiquitination (Fig 1C) [9,28]. Although the MCC represses APC/C activity towards cyclin B and securin, APC/C$^{MCC}$ is capable of ubiquitinating cyclin A [18] and MCC subunits [14,15,20]. Therefore, we tested whether APC/C$^{MCC}$ can ubiquitinate Nek2A. Addition of increasing concentrations of MCC to APC/C$^{Cdc20}$ strongly inhibited securin ubiquitination, but only minimally affected Nek2A ubiquitination (Fig 3A). Thus, either Nek2A out-competes MCC for binding to APC/C$^{Cdc20}$ or Nek2A is recognized by APC/C$^{MCC}$, allowing its ubiquitination in the presence of the MCC.

To understand the mechanism of recognition of Nek2A by the APC/C in the presence of the MCC, we analysed our newly improved, higher resolution APC/C$^{MCC}$ cryo-EM reconstruction (Figs EV2E and 4). As previously shown [14,15], this complex features the APC/C$^{MCC-closed}$ and APC/C$^{MCC-open}$ conformations. Using a newly released version of RELION [31], we reprocessed our previously published cryo-EM data set[10]. We refined both open and closed states of APC/C$^{MCC}$ to 3.8 Å resolution (Figs EV2E, EV4C–E and EV5). In APC/C$^{MCC-closed}$, the IR tail of Cdc20$^{MCC}$ is well defined with clear side-chain density, indicating that the IR motif is inserted into the Apc8A IR tail binding pocket (Figs 1G and EV4C). Conversely, in APC/C$^{MCC-open}$, the same motif density is much weaker, indicating dissociation of the IR tail from this binding site (Fig EV4D and E).

Based on these findings, we hypothesized that Nek2A would bind with higher affinity to the open state of APC/C$^{MCC}$ in which the Nek2A MR tail binding site on Apc8A would be disengaged from the Cdc20$^{MCC}$ IR tail. To test this, we used analytical size-exclusion chromatography to determine the amount of Nek2A bound to APC/C$^{MCC}$ and assessed the effect of mutations that stabilize either the open or the closed states. As shown in Figs 3B and EV1C, in comparison with the APC/C, Nek2A binding to APC/C$^{MCC}$ is reduced. Wild-type APC/C$^{MCC}$ exists mainly in the closed conformation [14]. In the context of a mutant of APC/C with the Apc15 subunit deleted (APC/C$^{ΔApc15-MCC}$), which locks APC/C$^{MCC}$ in the closed conformation [14,15], Nek2A binding is further reduced

(Figs 3C and EV1D). Notably, the Apc15 deletion does not inhibit Nek2A binding to apo-APC/C (Figs 3D and EV1D). Strikingly, the BubR1Wm mutant, which destabilizes the interaction between the MCC and the Apc2$^{WHB}$ domain, thereby promoting the open state [14], enhances Nek2A binding to APC/C$^{MCC}$ (Figs 3E and EV1E). Importantly, this effect is specific for Nek2A. Securin binds very weakly to both the APC/C$^{MCC}$ and the APC/C$^{MCC-BubR1Wm}$ mutant (Figs 3F and EV1E). As expected, deleting the IR tail of Cdc20$^{MCC}$, which binds the MR pocket 1 (MCCΔIR mutant), also stimulates Nek2A binding (Figs 3E and EV1E).

In conclusion, the state of the APC/C responsible for the recognition and ubiquitination of Nek2A in prometaphase is APC/C$^{MCC-open}$.

## Mechanism of Nek2A ubiquitination by APC/C$^{MCC}$

The two E2 enzymes UbcH10 and UbcH5 are responsible for ubiquitin chain initiation by the APC/C [5–7,32]. UbcH10 is the main E2 of the APC/C in mitosis [33]. Compared with most E2s, UbcH10 has an N-terminal extension, removal of which causes a checkpoint bypass [33]. The activity of UbcH10 with the APC/C requires the Apc2$^{WHB}$ domain, whereas the activity of UbcH5 does not [30]. The position of the WHB domain in the APC/C−Nek2A complex is strikingly different from that in structures of the APC/C−UbcH10 complexes with the coactivator-dependent substrate Hsl1 [27,30] (Fig 4A). These observations suggest that Nek2A may be more efficiently ubiquitinated by an APC/C−E2 complex that differs from the one preferred by the coactivator-dependent substrates. For example, cyclin A ubiquitination is more efficient with UbcH10 than with UbcH5 (Fig 4B) [34]. By contrast, Nek2A is more strongly ubiquitinated with UbcH5 than UbcH10 (Fig 4C). This indicates that, differing from cyclin A, which has a strong preference for UbcH10, Nek2A can use either E2. Potentially, this could contribute to the prometaphase-specific degradation of Nek2A because Nek2A, using two different E2, enzymes have a much higher chance of being ubiquitinated. To test the requirement of cyclin A2 and Nek2A for different E2 enzymes, we depleted either UbcH10 or UbcH5 in cells (Fig 4D) and monitored the stability of eGFP-cyclin A2 [18] and eGFP-Nek2A during mitosis (Fig 4E and F, Appendix Fig S1, Table EV2, Movies EV1–EV4, Movies EV5–EV7 and Source data Cells). Depletion of UbcH10 but not UbcH5 mildly delayed the degradation of eGFP-cyclin A2 (Fig 4F). However, degradation of eGFP-Nek2A was unaffected by depletion of either E2 enzyme. In agreement with our *in vitro* ubiquitination assays, this indicates that cyclin A2 prefers UbcH10 for its efficient ubiquitination by the APC/C, whereas Nek2A can be ubiquitinated by the APC/C in conjunction with either UbcH5 or UbcH10.

To confirm this result, we performed a double depletion of UbcH5 and UbcH10 together in cells expressing either eGFP-cyclin

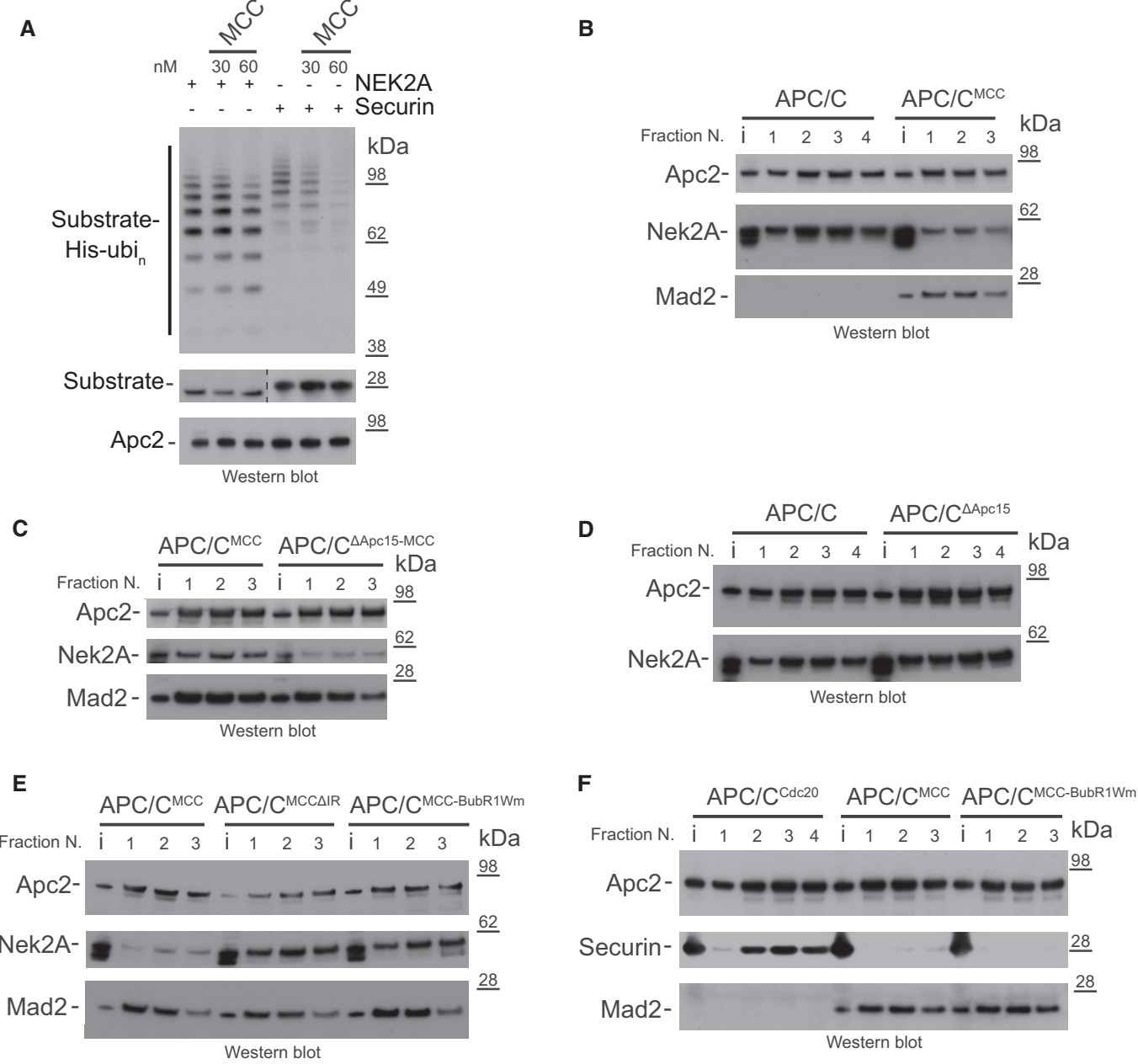

**Figure 3. Nek2A binds the APC/C^MCC complex when the APC/C^MCC closed conformation is destabilized.**

A  Ubiquitination reactions of either Nek2A or securin by the APC/C^Cdc20 and the effect of increasing concentrations of the MCC.

B–F  Size-exclusion chromatography peak fractions of APC/C complexes with either Nek2A (B-E) or securin (F). APC/C^ΔApc15 is the Apc15 deleted mutant APC/C, and MCCΔIR is the Cdc20 IR tail deleted mutant MCC. MCC-BubR1Wm is the MCC mutant at the APC2^WHB binding surface on BubR1 [14]. Chromatograms are shown in Fig EV1.

Source data are available online for this figure.

A2 or eGFP-Nek2A (Fig 4D–F, Movies EV5 and EV8 and Appendix Fig S1). Under this condition, degradation of eGFP-Cyclin A2 is strongly delayed, indicating that in cells, cyclin A2 might be able to use both E2 enzymes, but prefers UbcH10 since the effect of the single depletion of UbcH10 is much more pronounced than single depletion of UbcH5. Additionally, we observed that in the absence of both E2 enzymes, the degradation of Nek2A is now also

delayed, again confirming our *in vitro* assays. We noticed that even after depletion of the E2 enzymes in cells, the proteins were not completely absent (Fig 4D), which could account for the remaining degradation we observed.

In conclusion, the prometaphase-specific degradation of Nek2A depends on two distinct mechanisms. Firstly, Nek2A can bind the APC/C^MCC-open, and secondly, in contrast to cyclin A, Nek2A can

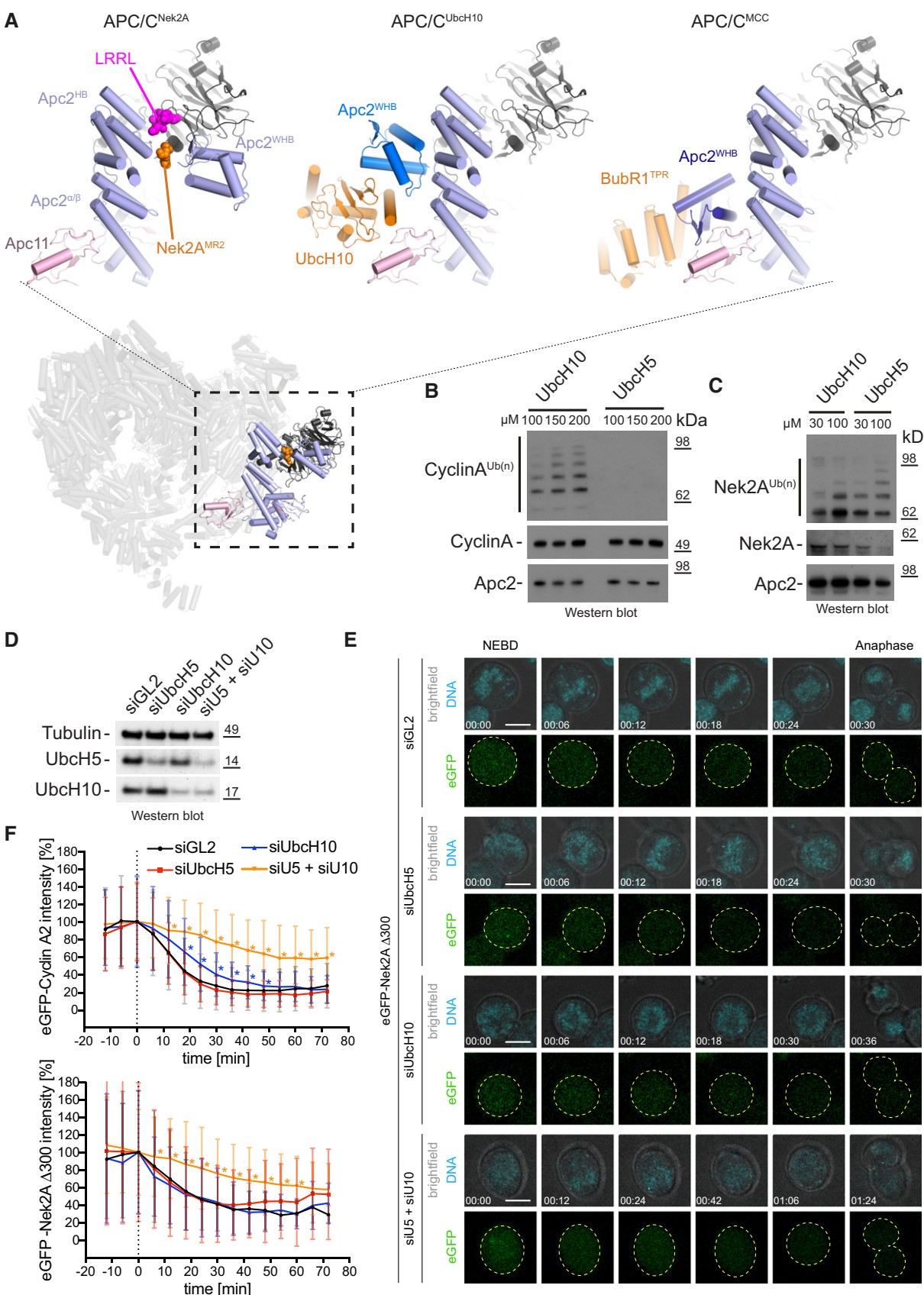

**Figure 4.**

**Figure 4.   Substrate-dependent E2 preferences in APC/C ubiquitination.**

A   Bottom: Overview of the APC/C–Nek2A structure is shown coloured as in Fig 2. Top: Close up of the Apc2 and Apc4 binding site for Nek2A in the APC/C–Nek2A structure (this study, left), APC/C$^{UbcH10\text{-}Hsl1}$ structure [27] and APC/C$^{MCC}$ structure [14].
B   Ubiquitination reactions of cyclin A by the APC/C$^{Cdc20}$ in the presence of increasing concentrations of either UbcH10 or UbcH5.
C   Ubiquitination reactions of Nek2A by the APC/C$^{Cdc20}$ in the presence of increasing concentrations of either UbcH10 or UbcH5.
D   Western blot of Hek293 cells showing the siRNA-mediated depletion of UbcH5, UbcH10 or both E2 enzymes together. Tubulin serves as a loading control.
E   Exemplary still images from time courses between NEBD and anaphase of eGFP-Nek2A degradation in HEK cells. Cells were either treated with siGL2 as control or depleted of the indicated E2 enzymes. The chromosomes are coloured in cyan and eGFP-Nek2A in green, with the outline of the cells are indicated with dashed yellow lines. Time is given as hh:mm. Scale bar 10 μm. See also Movies EV1–EV4 and Appendix Fig S1.
F   Degradation profiles of eGFP-cyclin A2 (top) and eGFP-Nek2A (bottom) in HEK cells during mitosis. The time point of NEBD is marked at 0 min in the graphs. Asterisks indicate values that are significantly different from the same time point of the siGL2 control as determined by a Mann–Whitney *U*-test (the statistics are listed in Table EV2). Mean ± SD is shown. The number of cells analysed are N = 66 (Cyclin A2 siGL2), 52 (Cyclin A2 siUbcH5), 28 (Cyclin A2 siUbcH10), 27 (Cyclin A2 siU5/siU10), 80 (Nek2A siGL2), 64 (Nek2A siUbcH5), 21 (Nek2A siUbcH10) and 32 (Nek2A siU5/siU10). All data are from at least two biological replicates. See also Movies EV5–EV8.

Source data are available online for this figure.

use efficiently both UbcH10 and UbcH5 (Fig 5). The latter mechanism would reduce the competition between Nek2A and cyclin A on the APC/C catalytic site as they have a different preference for APC/C-E2s complexes.

Our cryo-EM structure of the APC/C–Nek2A complex shows that Nek2A binding to the APC/C is strictly dependent on its C-terminal MR motif (Fig 1A and B). We found two MR tail binding sites. One MR tail binds the Apc8A TPR subunit, and the other binds a newly identified pocket formed by Apc2$^{CR3}$, Apc2$^{WHB}$ and Apc4$^{WD40}$. The

interaction with Apc8A is critical for optimal Nek2A binding to the APC/C [24], as the MCC, which binds to the same site through the Cdc20$^{MCC}$ IR tail, reduces Nek2A binding. Indeed, we show that removal of the Cdc20$^{MCC}$ IR tail strengthens the binding of Nek2A to APC/C$^{MCC}$ (Fig 3E).

Conversely, the binding site for the Nek2A MR tail on Apc2-Apc4 is not required for Nek2A binding affinity (Fig EV1F and Appendix Fig S2B–D), but it is required for efficient Nek2A ubiquitination (Fig 2C). Avidity effects between two MR tails and one

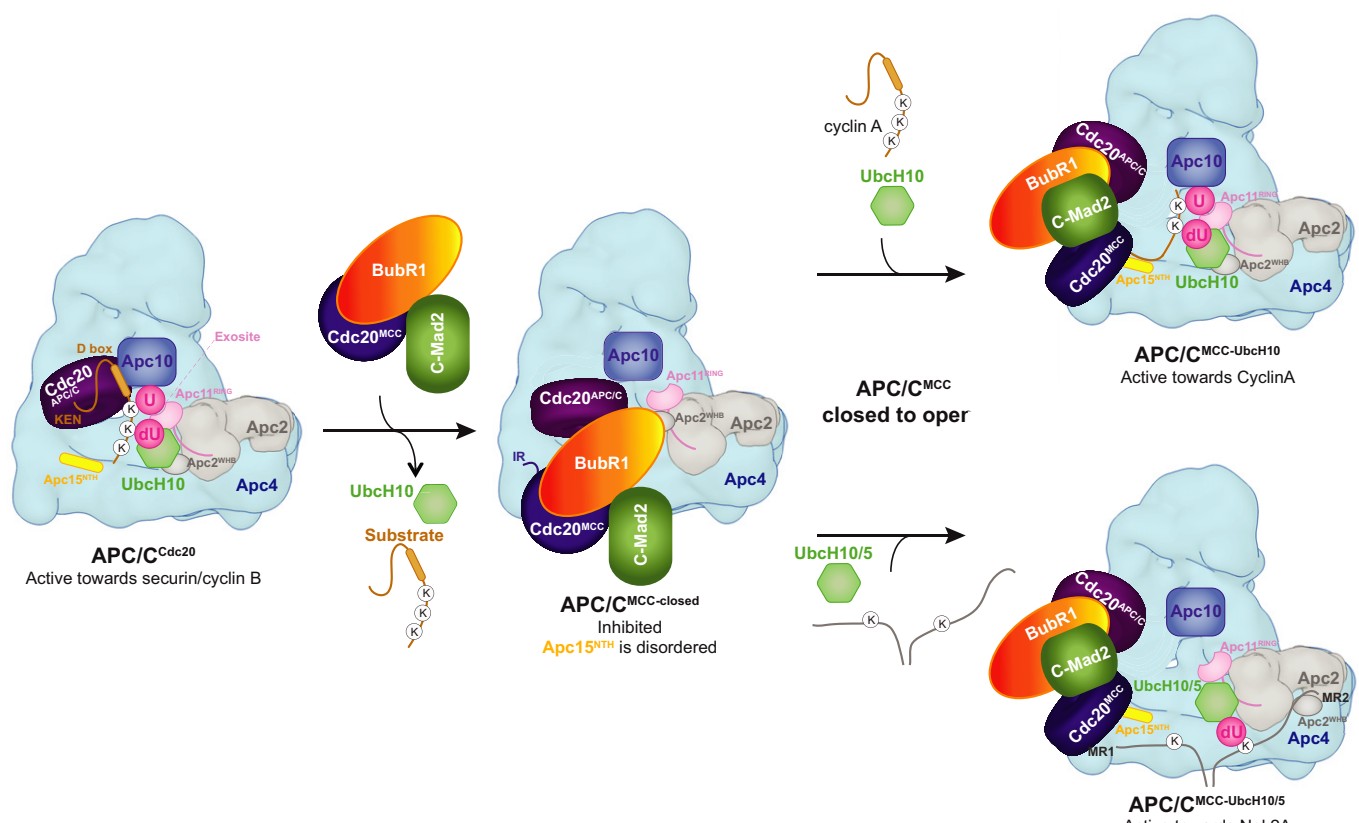

**Figure 5.   Model of Nek2A degradation during prometaphase.**
Cartoon illustrating the recognition of Nek2A by APC/C$^{MCC}$. KEN-box and D-box-dependent substrates such as securin and cyclin B do bind APC/C$^{MCC}$. In contrast, cyclin A and Nek2A bind APC/C$^{MCC\text{-}open}$. Cyclin A is ubiquitinated by UbcH10, and Nek2A has more chances of being ubiquitinated once bound to the APC/C$^{MCC\text{-}open}$ by using both UbcH10 and UbcH5. The positioning of UbcH5 with respect to the APC/C catalytic site is based on mutagenesis data [30,46].

binding site on Apc8 would allow high-affinity binding of Nek2A to the APC/C by decreasing the substrate off-rate [24]. How would the binding to the Apc2-Apc4 site additionally stimulate the ubiquitination of Nek2A? One possible explanation is that, in contrast with other well-characterized substrates such as geminin and cyclin B, the distance between the degron and the ubiquitinated lysine in Nek2A (Lys361) [35] is quite long. The distance between the Nek2A MR tail and Lys361 is 54 residues, compared with 18 and 13 residues for geminin and cyclin B, respectively. Therefore, a second binding site for the Nek2A dimer on the catalytic subunit Apc2 would restrain the range of conformations of the target lysine with respect to the APC/C catalytic site and therefore enhance Nek2A ubiquitination.

Part of the MR pocket 2 is Apc2$^{WHB}$ which, in the APC/C–Nek2A complex, is positioned in a configuration remote from its binding site with BubR1 of APC/C$^{MCC-closed}$. This corroborates that Nek2A is a selective binder of APC/C$^{MCC-open}$ where BubR1 no longer engages Apc2$^{WHB}$. Thus, Nek2A would promote APC/C$^{MCC-open}$ by first displacing the IR tail of Cdc20$^{MCC}$ from Apc8A and second by repositioning Apc2$^{WHB}$ away from BubR1. Previous work had shown that cyclin A also promotes APC/C$^{MCC-open}$, although through an undefined mechanism [18]. The APC/C−Nek2A structure shows that Nek2A induces a non-canonical conformation of the Apc2$^{WHB}$ domain. The WHB domain is positioned some 60 Å away from its canonical position observed in APC/C$^{MCC-closed}$ [14,15] and APC/C−UbcH10 complexes (Fig 4A) [27,30]. This arrangement suggests that once Nek2A binds to APC/C$^{MCC-open}$, Nek2A ubiquitination would be prioritized over the conversion to APC/C$^{MCC-closed}$. Importantly, by inducing the APC/C$^{MCC-open}$ state, Nek2A relieves MCC-mediated catalytic suppression of the APC/C, thereby allowing Nek2A ubiquitination.

MR pocket 2 is adjacent but not overlapping with the LRRL binding site for Ube2S [36] (Fig 4A), suggesting that there is no competition between Nek2A and Ube2S for APC/C binding. This is consistent with the finding that Ube2S is required for the efficient ubiquitination and degradation of Nek2A in early mitosis [37]. Importantly, mutations of MR pocket 2 do not affect Ube2S-mediated ubiquitin chain elongation (Appendix Fig S2A).

Previous work had shown that different substrates can be ubiquitinated with different efficiencies by the APC/C in conjunction with either UbcH10 or UbcH5 [34]. We show that in contrast to cyclin A, Nek2A is ubiquitinated more efficiently by UbcH5 relative to UbcH10. We also show in cells that in contrast to cyclin A, Nek2A is efficiently ubiquitinated by both UbcH10 and UbcH5 (Fig 4D–F).

In conclusion, our work sheds light on coactivator-independent substrate recognition, how Nek2A is ubiquitinated by APC/C$^{MCC}$, and on the role of multiple E2s in achieving APC/C cell cycle-dependent substrate specificity. This study also raises the possibility that these mechanisms may be exploited by yet undiscovered APC/C substrates.

# Materials and Methods

## Expression and purification of human APC/C, MCC and coactivators

The genes for recombinant human APC/C were cloned into a modified MultiBac system, expressed and purified as described [38,39]. Human Cdc20 and MCC were expressed and purified as described [14].

## Cloning, expression and purification of APC/C substrates

Nek2A residues 301–406, wild type and mutants were cloned into a pETM41 vector with an N-terminal His-MBP tag. The proteins were expressed in B834 (DE3) pLysS cells at 18°C overnight. Cell pellets were lysed in lysis buffer (50 mM Tris–HCl pH 8.0, 500 mM NaCl, 5% glycerol, 15 mM imidazole and 0.5 mM TCEP) supplemented with 0.1 mM PMSF, lysozyme, 5 units/ml benzonase and CompleteTM EDTA-free protease inhibitors (Roche). After sonication, the cells were centrifuged at 48,000×$g$ for 1 h at 4°C and the supernatant was incubated with Ni-NTA agarose (Qiagen) resin for 1 h at 4°C. The resin was washed with lysis buffer, and the protein was eluted in elution buffer (50 mM Tris–HCl pH 8.0, 500 mM NaCl, 5% glycerol, 250 mM imidazole and 0.5 mM TCEP). The imidazole was removed by dialysis of the sample in dialysis buffer (50 mM Tris–HCl pH 8.0, 250 mM NaCl, 5% glycerol and 0.5 mM TCEP). The MBP-fused version was used for ubiquitination assays and binding studies. For the ubiquitination assay in Fig 3, the protein was cleaved by incubating with TEV overnight during the dialysis step. The protein was further purified with a 6 ml Resource Q column and gel filtration on a Superdex 75 10/300 column.

Full-length human securin was tagged with an N-terminal GST-tag by cloning into a pGEX vector. The protein was expressed in BL21 (DE3) Star cells at 18°C overnight. The cell pellets were lysed in lysis buffer, following sonication the cleared lysate was incubated with the glutathione sepharoseTM 4B (GE Healthcare) for 3 h at 4°C. The resin was washed with the lysis buffer, and the GST-tag of securin was cleaved off with 3C PreScission protease overnight at 4°C. The flow-through from the resins was collected and applied on a 6 ml Resource Q column, and the resulting protein was further purified by size-exclusion chromatography using a Superdex 75 16/60 column (GE Healthcare) in the gel filtration buffer.

Full-length cyclin A-Cdk2-Cks2 complex was expressed and purified as described [18].

## Binding assay of APC/C complexes with substrates

Ten micromoles of purified APC/C complex was mixed with 1.5 times excess (molar ratio) of purified MBP-Nek2A, securin, Cdc20 and MCC constructs as indicated in the experiments and incubated on ice for 15 min. The mixture was purified on a Superose 6 Increase 3.2/300 column using the Microakta system (GE Healthcare), and the eluted peak fractions were analysed by SDS–PAGE on 4–12% NuPAGE Bis-Tris gels.

## Ubiquitination assays

Ubiquitination assays were performed with 15 nM recombinant human APC/C, 30 nM UBA1, 150 nM UbcH10, 20 μM ubiquitin, 0.2 μM substrates, 5 mM ATP, 0.25 mg/ml BSA and 25 nM Cdc20 in a 15 μl reaction volume with 40 mM Hepes pH 8.0, 150 mM NaCl, 10 mM MgCl$_2$ and 0.6 mM DTT.

Reaction mixtures were incubated at room temperature for 30 min and terminated by adding SDS/PAGE loading dye. Reactions were analysed by 4–12% NuPAGE Bis-Tris gels followed by Western blotting with an antibody against the His-tag of ubiquitin to detect the His-tag of the ubiquitin-modified substrates (Clontech, mouse

monoclonal, 631212) and HRP-conjugated sheep anti-mouse antibody (GE Healthcare, NXA931V).

Detection of Nek2A was performed with the mouse monoclonal antibody (BD Transduction Laboratories, 610594). Detection of cyclin A2 was performed with the rabbit monoclonal cyclin A2 antibody (Abcam, ab32386), detection of securin with the rabbit monoclonal securin antibody (Invitrogen, 700791), detection of Apc4 with the rabbit monoclonal Apc4 antibody (Abcam, ab72149), detection of Apc2 with the rabbit monoclonal Apc2 antibody (Cell Signaling, 12301) and HRP-conjugated donkey anti-rabbit antibody (Thermo Fischer, SA1-200). Primary antibodies were used at a dilution of 1:1,000 and secondary antibodies at a dilution of 1:5,000.

## Cryo-electron microscopy

For cryo-EM, 2.5 μl aliquots of freshly purified APC/C complexes at ~0.15 mg/ml were applied onto Quantifoil R2/2 grids or R3.5/1 grids coated with a layer of continuous carbon film (~50 Å thick). Grids were treated with a 9:1 argon:oxygen plasma cleaner (Fischione Instruments Model 1070 Nano clean) for 20–40 s depending on the thickness of the carbon before use. The grids were incubated for 30 s at 4°C and 100% humidity before blotting for 5 s and plunging into liquid ethane using a FEI Vitrobot III. Two cryo-EM data sets were collected for the APC/C–Nek2A complex. One collected on a Tecnai Polara electron microscope with a pixel size of 1.36 Å pixel$^{-1}$ using the Falcon 3 detector in integration mode as described [14], and another collected on a Titan Krios electron microscope with a pixel size of 1.1 Å pixel$^{-1}$ using the Gatan K2 detector in electron counting mode. For the latter data set, micrographs were recorded with a defocus range of −0.5 to −3.0 μm. The exposure time for each micrograph was 7 s at a dose rate of 5 electrons/pixel/s, and 28 movie frames were recorded for each micrograph.

## Image processing

All movie frames were aligned by motioncor 2 [40] programme before subsequent processing. First, the contrast transfer function (CTF) parameters were calculated with Gctf [41]. Particles in 400 × 400 pixels were selected by automatic particle picking in RELION 3.0.8 [42]. The following steps were performed to exclude bad particles from the data set: (i) automatically picked particles in each micrograph were screened manually to remove ice contaminations; (ii) after particle sorting in RELION, particles with poor similarity to reference images were deleted; (iii) two-dimensional classification was performed, and particles in bad classes with poorly recognizable features were excluded. The remaining particles were refined using RELION three-dimensional (3D) refinement and divided into three classes using 3D classification in RELION with fine-angle search. During this process, leftover bad particles were removed. Beam-induced particle motion was corrected using Bayesian polishing in RELION 3.0.8. Particles were also subjected to CTF refinement and beam tilt correction as implemented in RELION 3.0.8. Particle subtraction was performed for the region of interest, either the substrate recognition site in Apc8A or the portions of Apc2 and Apc4 (Fig EV3), using a soft mask (edge 10) surrounding the region of interest followed by focused 3D classification. Classes

with improved substrate recognition sites were 3D-refined in RELION individually. Density of the Apc2$^{WHB}$ domain was further improved by the multi-body refinement protocol implemented in RELION 3.0.8 [31].

The data from [Ref. 14] were reprocessed with the same protocol for obtaining the improved resolution structures of APC/C$^{MCC}$. After 3D classification with local searches, APC/C$^{MCC\text{-}closed}$ was used for one round of 3D multi-body refinement. The proportion of APC/C$^{MCC\text{-}closed}$ and APC/C$^{MCC\text{-}open}$ differed from [Ref. 14] because only the very best particles, which showed better alignment accuracy, were included in this analysis.

Final maps were sharpened and filtered in RELION post-processing. Local resolution of the maps was estimated using RELION. All resolution estimations were based on the gold-standard Fourier shell correlation (FSC) calculations using the FSC = 0.143 criterion. A summary of all EM reconstructions obtained in this paper is listed in Table EV1.

## Tissue culture and generation of stable cell lines

Hek293 FlpIn-TRex cells (Invitrogen) were cultured in DMEM (Gibco) supplemented with 10% tetracycline-free FBS (PAN Biotech) at 37°C and 5% CO$_2$.

For the stable integration of eGFP-Nek2A-Δ300 into the genome, the gene was cloned into the pcDNA5-FRT-TO vector (Invitrogen) with an N-terminal eGFP-tag. Hek293 FlpIn-TREX cells were cotransfected with the pCDNA5-FRT-TO-Nek2A plasmid and pOG44, containing the flippase, using the HBS method. In short, cells were seeded the evening before transfection and the medium was exchanged the next morning. Both plasmids were mixed with 160 mM CaCl$_2$ and 2× HBS buffer (final concentrations: 137 mM NaCl, 5 mM KCl, 0.7 mM Na$_2$HPO$_4$, 7.5 mM D-glucose and 21 mM HEPES) and added to the cells. The next steps were performed according to the Invitrogen manual. Cells were selected using 100 μg/ml Hygromycin B gold (InvivoGen). The generation of Hek293 FlpIn-TRex eGFP-Cyclin A2 cell was described before [18]. All stable cell lines were always kept under selection.

## RNAi mediated protein depletion, live cell microscopy and quantification of fluorescence images

Cells were seeded at a density of 150,000 cells per well of a 24-well plate (Corning) 6 h before transfection with siRNA. RNAi was performed using RNAiMAX reagent (Invitrogen) according to the manual. For the depletion of UbcH5 and UbcH10, 30 nM and 20 nM siRNA oligo mix were used, respectively. All siRNA oligos were described before [43,44]. The antibody used to detect UbcH5 was a rabbit polyclonal (Novus, NBP2-20783), and the antibody against UbcH10 was a rabbit polyclonal (Cell Signaling, 14234). Twenty-four hours after transfection, cells were blocked in 2.5 mM thymidine (Sigma), and the block was released by washing three times with fresh media 16 h later. Live cell microscopy was performed as described before [18]. A SP8 confocal microscope (Leica) equipped with a heated environmental chamber, an argon laser, a 630 nm laser line and a 40×/1.1 numerical aperture water immersion lens was used for imaging. The quantification of the resulting movies was also described before [18,45].

## Data availability

EM maps are deposited with EMDB with accession code EMD-10516 (https://www.emdataresource.org/EMD-10516) and EMD-10518 (https://www.emdataresource.org/EMD-10518). Protein coordinates are deposited with RCSB with PDB code 6TLJ (https://www.rcsb.org/structure/6TLJ) and 6TM5 (https://www.rcsb.org/structure/6TM5).

**Expanded View** for this article is available online.

## Acknowledgements

This work was supported by the Medical Research Council (MC_UP_1201/6) and a Cancer Research UK grant (C576/A14109) to D.B. and a Long Term and an Advanced EMBO Fellowships to C.A. We thank members of the Barford group for discussions; J. Yang and Z. Zhang for their help with the baculoviruses containing the APC/C genes. T. Nakane and S. Scheres for their help with RELION; G. Sharov, G. Cannone and S. Chen for EM facilities; J. Grimmett and T. Darling for computing. We thank S. Zhang for providing E1, E2, ubiquitin and cyclin A complexes for ubiquitination assays. We thank M. Gersch for providing UbcH5 for ubiquitination assays.

## Author contributions

CA cloned, purified proteins and performed the protein complex reconstitutions and biochemical analysis. CA prepared grids, collected and analysed EM data and determined the 3D reconstructions. CA fitted coordinates, built models and made the figures. TT performed live cell microscopy and *in vivo* data analysis. DB and CA directed the project and designed experiments. CA wrote the manuscript with inputs from DB and TT.

## Conflict of interest

The authors declare that they have no conflict of interest.

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
