## [Review Process File · EMBO Reports]

A unique binding mode of Nek2A to the APC/C allows its ubiquitination during prometaphase.

Claudio Alfieri, Thomas Tischer and David Barford

Review timeline:

Submission date:	8 December 2019
Editorial Decision:	7 January 2020
Revision received:	31 January 2020
Editorial Decision:	4 March 2020
Revision received:	11 March 2020
Accepted:	17 March 2020

Transaction Report:

1st Editorial Decision

7 January 2020

Thank you for the submission of your research manuscript to our journal. We have now received the full set of referee reports that is copied below.

As you will see, the referees acknowledge the interest and quality of your study and support publication after minor revision.

Given these constructive comments, we would like to invite you to revise your manuscript with the understanding that the referee concerns must be fully addressed and their suggestions taken on board. Please address all referee concerns in a complete point-by-point response.

Revised manuscripts should be submitted within three months of a request for revision; they will otherwise be treated as new submissions. Please contact us if a 3-months time frame is not sufficient for the revisions so that we can discuss the revisions further.

- 1) a .docx formatted version of the manuscript text (including legends for main figures, EV figures and tables). Please make sure that the changes are highlighted and clearly visible.
- 2) individual production quality figure files as .eps, .tif, .jpg (one file per figure). Please download our Figure Preparation Guidelines (figure preparation pdf) from our Author Guidelines pages <https://www.embopress.org/page/journal/14693178/authorguide> for more info on how to prepare your figures.
- 3) a .docx formatted letter INCLUDING the reviewers' reports and your detailed point-by-point

responses to their comments. As part of the EMBO Press transparent editorial process, the point-by-point response is part of the Review Process File (RPF), which will be published alongside your paper.

4) a complete author checklist, which you can download from our author guidelines (<<https://www.embopress.org/page/journal/14693178/authorguide>>). Please insert information in the checklist that is also reflected in the manuscript. The completed author checklist will also be part of the RPF.

5) Please note that all corresponding authors are required to supply an ORCID ID for their name upon submission of a revised manuscript (<<https://orcid.org/>>). Please find instructions on how to link your ORCID ID to your account in our manuscript tracking system in our Author guidelines (<<https://www.embopress.org/page/journal/14693178/authorguide#authorshipguidelines>>)

6) We replaced Supplementary Information with Expanded View (EV) Figures and Tables that are collapsible/expandable online. A maximum of 5 EV Figures can be typeset. EV Figures should be cited as 'Figure EV1, Figure EV2' etc... in the text and their respective legends should be included in the main text after the legends of regular figures.

7) Please update your Data availability section to follow the model below:

The accession numbers and database should be listed in a formal "Data Availability " section (placed after Materials & Method) that follows the model below (see also <<https://www.embopress.org/page/journal/14693178/authorguide#dataavailability>>). Please note that the Data Availability Section is restricted to new primary data that are part of this study.

Data availability

8) References: Please update the references to the numbered format of EMBO reports. The abbreviation 'et al' should be used if more than 10 authors. You can download the respective EndNote file from our Guide to Authors
<https://drive.google.com/file/d/0BxFM9n2IEE5oOHM4d2xEbmxN2c/view>

9) Unless the revision results in more than 5 main figures, your manuscript will be published in the "Reports" section. Therefore, please combine the Results and Discussion section. The character limit for Reports is 25,000 plus/minus 2,000 characters (excluding materials & methods and references).

10) Supplementary Table is difficult to read in its current form and could be submitted as Table EV1 (in .xls format).

11) We would also encourage you to include the source data for figure panels that show essential data. Numerical data should be provided as individual .xls or .csv files (including a tab describing the data). For blots or microscopy, uncropped images should be submitted (using a zip archive if multiple images need to be supplied for one panel). Additional information on source data and instruction on how to label the files are available

<<https://www.embopress.org/page/journal/14693178/authorguide#sourcedata>>.

12) Our journal encourages inclusion of *data citations in the reference list* to directly cite datasets that were re-used and obtained from public databases. Data citations in the article text are distinct from normal bibliographical citations and should directly link to the database records from which the data can be accessed. In the main text, data citations are formatted as follows: "Data ref: Smith et al, 2001" or "Data ref: NCBI Sequence Read Archive PRJNA342805, 2017". In the Reference list, data citations must be labeled with "[DATASET]". A data reference must provide the database name, accession number/identifiers and a resolvable link to the landing page from which the data can be accessed at the end of the reference. Further instructions are available at <<https://www.embopress.org/page/journal/14693178/authorguide#referencesformat>>.

13) Regarding data quantification:

- Please ensure to specify the name of the statistical test used to generate error bars and P values, the number (n) of independent experiments underlying each data point (not replicate measures of one sample), and the test used to calculate p-values in each figure legend. Discussion of statistical methodology can be reported in the materials and methods section, but figure legends should contain a basic description of n, P and the test applied.

IMPORTANT: Please note that error bars and statistical comparisons may only be applied to data obtained from at least three independent biological replicates. If the data rely on a smaller number of replicates, scatter blots showing individual data points are recommended.

- Graphs must include a description of the bars and the error bars (s.d., s.e.m.).

14) EMBO reports papers are accompanied online by A) a short (1-2 sentences) summary of the findings and their significance, B) 2-3 bullet points highlighting key results and C) a synopsis image that is 550x200-400 pixels large (width x height). You can either show a model or key data in the synopsis image. Please note that the size is rather small and that text needs to be readable at the final size. Please send us this information along with the revised manuscript.

15) As part of the EMBO publication's Transparent Editorial Process, EMBO reports publishes online a Review Process File to accompany accepted manuscripts. This File will be published in conjunction with your paper and will include the referee reports, your point-by-point response and all pertinent correspondence relating to the manuscript.

I look forward to seeing a revised version of your manuscript when it is ready. Please let me know if you have questions or comments regarding the revision.

REFeree REPORTS

Referee #1:

This paper explores how the protein Nek2A is degraded by the anaphase-promoting

complex/cyclosome (APC/C) during prometaphase. Nek2A is an important mitotic kinase and is one of the few substrates that get degraded during an active spindle assembly checkpoint (SAC). Understanding the mechanism of Nek2A degradation is therefore important as it provides insight into substrate recognition by the APC/C.

By combining elegant structural, biochemical and cell based assays this paper now provides an important step forward in understanding Nek2A degradation. The authors show by structural and functional studies that the two MR motifs of the Nek2A dimer engages two distinct binding pockets of the APC/C. Furthermore they show that Nek2A binds to the form of APC/C-MCC that is in the open conformation and that Nek2A can use both UbcH5 and UbcH10 as E2 enzymes.

Overall this is a really compelling story and I strongly recommend publication in EMBO reports with only minor corrections. There are few points that the authors can maybe touch upon in the text which I thought could be relevant.

- 1) Could the authors comment on why the Cdc20 IR motif is not bound to the new pocket they identify for the Nek2A MR motif (the pocket involving APC2).
- 2) Strictly speaking I guess the authors cannot know if the MR densities are coming from a single Nek2A dimer or from 2 Nek2A monomers or?
- 3) Have the authors conducted any statistical analysis of the live cell degradation assays to support their claims? I think the effects they are mentioning are clear but are they significant?
- 4) In their model in figure 5 they have positioned Ubc5 at the same position as UbcH10 but I am not sure there is any evidence for this. As Nek2A can use both E2s I guess there does not have to be a strict positioning of the lysines that become ubiquitinated in Nek2A.

Jakob Nilsson

Referee #2:

The authors present a very nice, efficient, analysis of how the APC/C can recognize the Nek2A protein kinase even when the spindle assembly checkpoint is active. An active checkpoint prevents the ubiquitination by the APC/C of nearly all of its substrates by promoting the binding of the "MCC" to the APC/C. Two substrates are notable for their escape of this inhibition, cyclin A and Nek2A. Cyclin A outcompetes the MCC for binding to the APC/C. The authors confirm and substantially extend previous reports that Nek2A binds the APC/C in a novel manner, bypassing the usual recruitment to APC/C activators such as Cdc20 via Destruction boxes and KEN boxes. Instead, the main recognition of Nek2A occurs through its C-terminal MR motif, related to the recognition of APC/C activators via their C-terminal IR motifs. Nek2A binds preferentially to the "open" conformation of APC/C(MCC), which can catalyze its ubiquitination. The authors reach these conclusions using predominantly a nice combination of structural analysis with rigorous and quite clean biochemical assays. This is an excellent study with few areas for improvement.

Minor comments:

- 1) It would be helpful for non-experts for the authors to indicate early on the composition of the MCC and of APC/C(MCC), particularly the presence of the two Cdc20 molecules. Readers may otherwise get confused and think that Nek2A can be ubiquitinated in the absence of an APC/C activator. Similarly, the authors should define terms such as Cdc20(MCC). The authors should check for any other terms that might confuse non-expert readers.
- 2) p. 10 near bottom/Fig. 4F. The delay in cyclin A degradation by knockdown of UbcH10 is really quite modest, especially given the very strong delay after double knockdown of UbcH10 and UbcH5. The authors should consider toning down their interpretation. Similarly, is the effect of UbcH10 knockdown on cyclin A actually statistically significantly different from the lack of effect on Nek2A degradation?
- 3) p. 17. The legends for Figs. 4B and 4C are swapped.
- 4) p. 17. More detail in the Fig. 3B-F legend would be helpful. For instance, descriptions of the

mutants, referral to the Supplement for the SEC samples....

- 5) Fig. 4E legend. Please replace "destruction" with "degradation" as used elsewhere.
- 6) p. 18. I didn't see a panel G in Supplemental Fig. 1. I also didn't see a label for the gel samples below panel F. Should that be part of G?
- 7) Supplemental Fig. 2D and 2E. The legends under the graphs are much too small. Perhaps they could be enlarged and presented to the right of the curves?
- 8) Supplementary Table 1. The text is much too small.

Referee #3:

The latest manuscript from Alfieri, Tischler, and Barford presents the structure and associated biochemistry and cell biology of APC/C-Cdc20 recognition of the substrate Nek2A. Nek2A stands out amongst APC/C substrates for its degradation during prometaphase requiring its MR-tail sequence, coiled-coil-induced dimerization and during a phase of the cell cycle when APC/C-Cdc20 is thought to be inhibited by binding to MCC. As expected for work from this group, the structural studies are high quality and maximized by state-of-the-art cryo EM refinement strategies. The studies support a model with the two MR tails in the dimer binding the APC8A-TPR subunit on one side and a novel pocket formed by a rearranged APC2-WHB domain WIT APC4-WDR40. The report also shows that UbcH5 is superior over UbcH10 at ubiquitinating Nek2A and provide a structural explanation for these findings. The new refinements of open and closed APC/C-MCC structures are a nice addition. It is clear that Nek2A ubiquitination is not completely simplistic, because this can involve UbcH10, which would reconfigure the APC2-WHB domain and eliminate one Nek2A binding site, and because the APC2-WHB/APC4-WD40 binding site for Nek2A apparently contributes to ubiquitination efficiency and not to affinity. Still, the authors' data explain many outstanding questions in the cell cycle and APC/C field. The manuscript is very well-written and beautifully explains complicated structural and biochemical results in a concise and clear way. The study presents much high-quality data. The work is novel, important, and of the highest quality, and I recommend rapid publication in EMBO Reports after minor changes to the text and clarification of the methods:

1. Some referencing seems incomplete. The authors should reference Gmachl et al. (2000) and Levenson et al. (2000) as showing UbcH5 functions with APC/C. Foster and Morgan (2012) and Uzunova et al (2012) should be cited alongside reference 11. There is no question that these authors made major breakthrough discoveries in their reference 12, but reference 16 should be cited alongside this.
2. Can the authors explain the use of low salt in their ubiquitination assays. Are some of the interactions electrostatic and screened by salt?
3. The authors state that the second MR-tail binding site does not overlap with that of UBE2S. Did the authors test the effects of mutations at this site on ubiquitination with UBE2S?
4. What antibody was used to detect APC2?

1st Revision - authors' response

31 January 2020

Referee #1:

- 1) Could the authors comment on why the Cdc20 IR motif is not bound to the new pocket they identify for the Nek2A MR motif (the pocket involving APC2).

Thank you for raising this important point. We added a new figure (Appendix Fig S2B-D). As shown in the conservation analysis (panel B), the IR tail motifs of the APC/C coactivators Cdc20 and Cdh1, comprise two main elements: (i) a hydrophobic residue (ϕ) at position -5 and the (ϕ)-arginine dipeptide at positions -1 and 0, respectively. The (ϕ) at -5 docks against a hydrophobic residue forming the coactivator binding site of Apc3A (panel C). Conversely, the (ϕ) at -5 is missing at the C-terminus of Nek2A (panel B).

Modelling of the Cdc20 MR tail onto the Nek2A MR pocket 2 (panel D), shows that (ϕ) in position -5 is unable to establish hydrophobic contacts with the APC/C.

Therefore, the Cdc20 IR motif would preferentially bind to the higher affinity IR tail binding site, which is on Apc3A.

- 2) Strictly speaking I guess the authors cannot know if the MR densities are coming from a single Nek2A dimer or from 2 Nek2A monomers or?

This statement is correct. From our cryo-EM reconstruction we cannot know if the MR densities are coming from a single Nek2A dimer. But we would expect that this is the case in light of previous biochemical data [24]: “Because dimerization through its LZ is essential for Nek2A degradation kinetics [24]” (page 6). This is also supported by our observation that mutating MR pocket 2 reduces Nek2A ubiquitination (Fig 2C) while Nek2A can still bind the APC/C (Fig EV1F and G).

- 3) Have the authors conducted any statistical analysis of the live cell degradation assays to support their claims? I think the effects they are mentioning are clear but are they significant?

Yes, we performed a statistical analysis of our live cell degradation assay, which is now included in Fig 4F and Table EV2. We also added the following sentence in the Fig 4F legend: “Asterisks indicate values that are significantly different from the same time point of the siGL2 control as determined by a Mann-Whitney U-test (the statistics are listed in Table EV2)”, (page 16).

- 4) In their model in figure 5 they have positioned Ubc5 at the same position as UbcH10 but I am not sure there is any evidence for this. As Nek2A can use both E2s I guess there does not have to be a strict positioning of the lysines that become ubiquitinated in Nek2A.

Based on mutagenesis data in REFs [30,38]. UbcH5 and UbcH10 require similar interactions with the Apc11 RING domain. We added the following sentence to the legend of Fig 5 (page 16): “The positioning of UbcH5 with respect to the APC/C catalytic site is based on mutagenesis data [30,38].”

Referee #2:

- 1) It would be helpful for non-experts for the authors to indicate early on the composition of the MCC and of APC/C(MCC), particularly the presence of the two Cdc20 molecules. Readers may otherwise get confused and think that Nek2A can be ubiquitinated in the absence of an APC/C activator. Similarly, the authors should define terms such as Cdc20(MCC). The authors should check for any other terms that might confuse non-expert readers.

We thank the reviewer for this suggestion. We added the following sentence in the Introduction of our manuscript (page 2): “The MCC is composed of Cdc20, Mad2, BubR1 and Bub3. The MCC and APC/C^{Cdc20} associate to form the APC/C^{MCC} complex which therefore contains two molecules of Cdc20, one is the APC/C coactivator (Cdc20^{APC/C}) and the other is Cdc20 of the MCC (Cdc20^{MCC}) [17].”

- 2) p. 10 near bottom/Fig. 4F. The delay in cyclin A degradation by knockdown of UbcH10 is really quite modest, especially given the very strong delay after double knockdown of UbcH10 and UbcH5. The authors should consider toning down their interpretation. Similarly, is the effect of UbcH10 knockdown on cyclin A actually statistically significantly different from the lack of effect on Nek2A degradation?

We agree that the delay in cyclin A degradation by knockdown of UbcH10 is modest in comparison to the double knockdown of UbcH10 and UbcH5. This could be due to a synergistic effect caused by the double depletion of two E2s, with partially redundant function.

However, the delay in cyclin A degradation by knockdown of UbcH10 is indeed significant. We performed a statistical analysis of our live cell degradation assay, which is now included in Fig 4F and Table EV2. We also added the following sentence in the Fig 4F legend: “Asterisks indicate values that are significantly different from the same time point of the siGL2 control as determined by a Mann-Whitney U-test (the statistics are listed in Table EV2)”, (page 16). As suggested, we toned down our findings by changing the following sentences: “this indicates that cyclin A2 prefers UbcH10 for its efficient ubiquitination by the APC/C”; “cyclin A2 might be able to use both E2 enzymes, but prefers UbcH10” (page 10).

- 3) p. 17. The legends for Figs. 4B and 4C are swapped.

We corrected this (page 15), thank you.

- 4) p. 17. More detail in the Fig. 3B-F legend would be helpful. For instance, descriptions of the mutants, referral to the Supplement for the SEC samples....

We added the following sentence to the legend of Fig 3B-F (page 15): “APC/CΔApc15 is the Apc15 deleted mutant APC/C, MCCAIR is the Cdc20 IR tail deleted mutant MCC. MCC-BubR1Wm is the MCC mutant at the APC2WHB binding surface on BubR1 [14]. Chromatograms are shown in Fig EV1.”

- 5) Fig. 4E legend. Please replace "destruction" with "degradation" as used elsewhere.

We corrected this (page 16), thank you.

- 6) p. 18. I didn't see a panel G in Supplemental Fig. 1. I also didn't see a label for the gel samples below panel F. Should that be part of G?

We corrected the legend now in Fig EV1 (page 17) panel F as follow: “F-G. Size exclusion chromatography chromatograms (F) and SDS-PAGE gels (G) of either APC/C wild type or 2/4m complexes with Nek2A.”

- 7) Supplemental Fig. 2D and 2E. The legends under the graphs are much too small. Perhaps they could be enlarged and presented to the right of the curves?

We corrected this, thank you.

- 8) Supplementary Table 1. The text is much too small.

We now show this data in .xls format as Table EV1.

Referee #3:

1. Some referencing seems incomplete. The authors should reference Gmachl et al. (2000) and Leverson et al. (2000) as showing UBCH5 functions with APC/C. Foster and Morgan (2012) and Uzunova et al (2012) should be cited alongside reference 11.

We added Gmachl et al. (2000) and Leverson et al. (2000) in page 2 [6] and [7],
“Foster and Morgan (2012) and Uzunova et al (2012)” in page 3 [20] and [22].

There is no question that these authors made major breakthrough discoveries in their reference 12, but reference 16 should be cited alongside this.

We agree with this statement, therefore, we added Yamaguchi et al. (2016) now [15] alongside Alfieri et al. (2016) [14] unless we refer to data from only Alfieri et al. (2016) in page 2, 3, 7, 8 and 12.

2. Can the authors explain the use of low salt in their ubiquitination assays. Are some of the interactions electrostatic and screened by salt?

Thank you for spotting this lack of clarity. We always use 150 mM of NaCl, which is now added in the Materials and Methods section (page 22).

3. The authors state that the second MR-tail binding site does not overlap with that of UBE2S. Did the authors test the effects of mutations at this site on ubiquitination with UBE2S?

The reviewer raises a very important point. In order to address this point we tested the effects of mutations of MR pocket 2 on ubiquitination with Ube2S (Appendix Figure S2A) and added the sentence “Importantly mutations of MR pocket 2 do not affect Ube2S-mediated ubiquitin chain elongation (Appendix Fig S2A).” in the main text (page 12).

4. What antibody was used to detect APC2?

We are sorry for this oversight and added the information in the material and methods section (page 23): “detection of Apc2 with the rabbit monoclonal Apc2 antibody (Cell Signaling, 12301)”.

2nd Editorial Decision

4 March 2020

Thank you for your patience while we have editorially reviewed your revised manuscript and please accept my apologies for the delay. I am now writing with an 'accept in principle' decision, which means that I will be happy to accept your manuscript for publication once a few minor issues/corrections have been addressed, as follows.

- 1) Please provide up to five keywords.

- 2) Please note that per editorial policy we do not allow "data not shown" (page 13, Figure 1C legend). Please either provide the data or remove the respective statement.

- 3) Movies:

Please remove the movie legends from the article file and provide them as simple README.txt file

- (one file per legend). Then zip each movie with its legend and upload the ZIP file. Please change the nomenclature to Movie EV_x and update the callouts in the text (main and Appendix).
- 4) Please add a callout to the movies also in the main text in addition to the callout you currently have in the legend for Figure 4.
 - 5) Please also add a callout to Fig. EV5 where relevant.
 - 6) I attach to this email a related manuscript file with comments by our data editors. Please address all comments and upload a revised file with tracked changes with your final manuscript submission. Please note that this document corresponds to an earlier version of your manuscript and lacks e.g. the information you added later on the number of replicates in Figure 4F.
 - 7) Appendix Figure S1B: please specify the number of samples analysed and whether technical or biological replicates were used. Please also define the error bars.
 - 8) Table EV2: please add the label "Table EV2" to the legend within the table.
 - 9) Please rearrange the panels of Figures EV3 and EV4 so that the figure is a true portrait format.
 - 10) Source data:
Please provide the source data for Figure 4E-F and S1 in separate .xls files for each figure and please also combine the other source data files to one file per figure (can also be zipped together).
 - 11) I attach here the synopsis image in its final size (550 pixels width). The text in the image is very difficult to read and should be adjusted.

Once you have made these minor revisions, please use the following link to submit your corrected manuscript:

<https://embor.msubmit.net/cgi-bin/main.plex?el=A6Ij1BZs3A3DADV7J6A9ftdAI2uyAhHMTlguopuVyJQwQY>

If all remaining corrections have been attended to, you will then receive an official decision letter from the journal accepting your manuscript for publication in the next available issue of EMBO reports. This letter will also include details of the further steps you need to take for the prompt inclusion of your manuscript in our next available issue.

Thank you for your contribution to EMBO reports.

2nd Revision - authors' response

11 March 2020

The authors performed all minor editorial changes.

Corresponding Author Name: Claudio Alfieri

Manuscript Number: EMBOR-2019-49831V2